# Improving the production of baculovirus expression vector by overexpression of IE0/IE1 through tandem promoter

**Sijun He[1], Weining Li[1], Ruirui Zhang[1], Hao Nan[1], Wangcheng Song[1,2], Xiaodong Xu[1],**

**1** College of Life Sciences, Northwest A&F University, Yangling, Shaanxi, China, **2** Shaanxi Sky Pet Biotechnology Co., Ltd, Xi'an, Shaanxi, China

\* xuxd@nwsuaf.edu.cn

## Abstract

The baculovirus expression vector system, known for its protein production in insect cells and has been criticized for its relatively low expression capacity. IE0/IE1, acknowledged vital early regulators of baculovirus, are indispensable for the virus proliferation and regulate the expression of various genes within the virus. Prior research has reported a substantial rise in exogenous gene expression upon overexpression of IE01. In this study, to mitigate the risk of generating defective viruses due to homologous recombination, we introduced an additional promoter in vivo within the viral genome, thus overexpressing IE0/IE1. The research outcomes demonstrate that the expression of exogenous proteins is notably enhanced without the homologous regions sequence for enhancement. In parallel, they still indicate that the upregulation of IE0/IE1 not only boosts viral titers but also enhances apoptosis within cellular populations. In sum, we successfully constructed a novel baculovirus expression vector that significantly enhances the expression of exogenous genes, presenting a new perspective for optimizing the baculovirus expression vector system.

## Introduction

Baculoviruses are a class of double-stranded DNA viruses that predominantly infect insects, particularly those belonging to the Lepidoptera order [1]. Among all baculoviruses, *Autographa californica* multiple nucleopolyhedrovirus (AcMNPV) is the most commonly used and has served as a model baculovirus in virological studies [2]. Due to their unique biological characteristics and safety profile, baculoviruses are widely used as tools for expressing foreign genes in the field of molecular biology and biotechnology, especially for the production of recombinant proteins [3]. Since the first successful expression of human beta interferon using the baculovirus expression vector system (BEVS) in 1983, both the industrial and laboratory have utilized the BEVS to express lots of recombinant proteins, and this practice continues until today [4]. Compared to other protein expression systems, such as bacteria, yeast, and animals, BEVS offers several advantages. Even though their protein production trails that of bacteria, BEVS expressed proteins exhibit unique biological activity thanks to correct folding and post-translational modifications (PTM) [5,6]. In contrast to mammalian expression

**Data availability statement:** All datas are contained within the paper.

**Funding:** This study was financially supported by two grants: 1. Shaanxi Province Key Research and Development Program (Project No. 2022QCY-LL-52), recipient: Hao Nan 2. A collaborative project with Shaanxi Sky Pet Biotechnology Co., Ltd. (Project No. K4050722064), recipient: Wangcheng Song The funders had no role in study design, data collection and analysis, decision to publish, or preparation of the manuscript.

**Competing interests:** The authors have declared that no competing interests exist.

systems, BEVS provides lower costs and enhanced safety. Over the past four decades, BEVS has been widely applied in industrial production. Meanwhile, researchers have continuously worked to optimize the BEVS to enhance the expression of exogenous genes.

To get superior protein expression vectors, scientists have conducted extensive modifications and optimizations on the BEVS [7]. Especially, gene editing technology has played a significant role in optimizing the BEVS. By deleting nearly 10 kb of non-essential genes for the baculovirus, the unnecessary genetic burden of the recombinant virus genome is reduced, leading to increased exogenous protein production and secretion efficiency [8]. Additionally, the absence of chitinase (*chiA*) and cathepsin (*v-cath*) improves protein secretion efficiency of the BEVS and reduces degradation of recombinant proteins [9,10]. PTM typically occur after protein translation and encompass processes such as phosphorylation, glycosylation, ubiquitination, and acetylation. These modifications are essential for the normal biological functions of recombinant proteins [11]. Unlike mammalian cells, insect cells are unable to add terminal galactose and sialic acid to the recombinant proteins due to the lack of glycogen or glycosidase activity. Consequently, the BEVS cannot synthesize mammalian-type glycan proteins. However, by introducing additional enzymes related to glycosylation and generating transgenic cell lines, the glycosylation proteins of insect cell can be made more similar to that of mammalian cells, refer to this review to get more details [12]. Insect cells resist the invasion of baculovirus by activating the apoptosis pathway. Therefore, prolonging the survival time of insect cells can significantly enhance the efficiency of the BEVS. After using shRNA to specifically target the *caspase-1* of insect cells, it was discovered that it can effectively delay apoptosis in host cells and prolong the duration of exogenous gene expression [13]. Usually, viral late promoters are used for the expression of exogenous genes, but they only start expressing from the 22nd hour after viral infection until the host dies. Therefore, enhancing the expression level of late promoters and prolonging the survival time of the host cells are currently the challenges faced by BEVS optimization.

There are five immediate early genes, including *ie0, ie1, ie2, me53,* and *pe38*, they all have been reported to have *trans*-activator functions [14]. AcMNPV IE1 is a 66.9 kDa protein, serving as an important regulatory factor involved in the replication and transcription of the entire life cycle of baculovirus. The N-terminus of IE1 contains two acidic domains separated by a basic domain I that is necessary for *hr* (homologous region) binding [15,16]. Located in the C-terminus are basic domain II and a helix-loop-helix domain, which are all involved in DNA binding, nuclear import, dimerization, and other processes [17,18]. AcMNPV IE0 is a 72.6 kDa protein, being the only known baculovirus spliced gene that produces an alternate protein product [19]. IE0 comprises 636 amino acids, including 38 amino acids encoded by *orf141*, 16 amino acids encoded by the upstream sequence of the untranslated region of IE1, and 582 amino acids from entire IE1 protein. Therefore, except for the first 54 amino acid residues at the N-terminus, IE0 and IE1 have the same remaining sequences. *ie0 and ie1* stimulate transcription by forming homodimers or heterodimers and their ability to transcribe is enhanced with the binding to *hr* [20,21]. IE0 mainly exists in the early stage of viral infection, peaking at 4 hours post infection (hpi), and then decreasing, while IE1 mainly exists in the later stage of viral infection. After knocking out both *ie0 and ie1*, the Stewart team found that the virus was unable to replicate. However, when only IE0 or IE1 is expressed, the virus can recover its replication ability. When only IE0 is expressed, viral infection is significantly impaired. When only IE1 is expressed, the ability to produce budded virion (BV) is similar to that of the wild-type (WT), but the production of polyhedra is reduced. In short, the complex of IE0 and IE1 is the most favorable condition for viral replication and either IE0 or IE1 is indispensable for the replication of the virus [22]. A study by Huijskens reported that the ratio of IE0 and IE1 is different throughout the infection cycle, and different ratios also have

different effects on the activation of late genes. When IE0 and IE1 are both highly expressed, there seems to be an antagonistic effect, resulting in decreased expression of late genes, and only when they are expressed in a specific ratio can the expression of late-stage genes be maximally activated [23]. IE1 stimulates its own promoter and inhibits the expression of the *ie0* promoter. On the other hand, IE0 activates the *ie1* promoter but does not affect the expression of its own promoter [24]. As demonstrated by the transient replication assay, *Lymantria dispar* multiple nucleopolyhedrovirus (LdMNPV) IE0 stimulates the replication the reporter plasmid containing the replication origin *hr4* of LdMNPV [25]. Additionally, baculovirus DNA replication induces apoptosis [26]. Therefore, it is speculated that overstimulation of baculovirus DNA replication caused by overexpression of IE0 may reduce viral fitness by increasing apoptotic cell death in the host. Moreover, ectopic expression of IE1 in *Sf21* cells can lead to a low level of apoptosis [27]. By silencing *ie1* through RNAi, AcMNPV-mediated cell lysis, caspase activation, and proteolytic processing were blocked. The proapoptotic effect of IE1 has been confirmed using *Drosophila DL-1* cells [28]. In fact, the initiation of AcMNPV DNA synthesis coincides with the activation of caspase [29–31]. Therefore, as the essential gene for viral DNA replication, IE0/IE1 may directly or indirectly participate in triggering apoptosis.

To optimize the BEVS efficiency, we employed the late promoter *vp39* to construct a novel expression vector that overexpresses IE0/IE1. Our findings indicated that this approach effectively boosts the expression of exogenous genes while simultaneously accelerating viral DNA replication and host cell apoptosis, offering a novel perspective for improving the BEVS.

## Methods and materials

### Viruses and cell lines

*Sf9* cells were cultured in SFX insect medium (Thermo Scientific HyClone) supplemented with 1% fetal bovine serum (FBS) at 27°C. Bacmid BAC10:KO1629

(named as BacI in this study) was propagated in *E. coli* strain HS996 [32]. Plasmid pTriEx-GFP was stored in our laboratory.

### Construction of recombinant baculoviruses

The *vp39* promoter was knocked into the BacI by Red/ET-based recombination. Briefly, rpsL-amp cassette was amplified by PCR using primers AMP-F and AMP-R, which were comprised of a homologous arm for upstream/downstream sequence of IE0 start coden and the rpsL-amp cassette. The amplified fragment was transformed into HS996 competent cells containing the Red®/ET® plasmid pSC101-BAD-gbaA and bacmid BacI, and the rpsL-amp cassette was inserted into the position immediately upstream of the IE0 coding region. The resulting bacmid was named as BacI-rpsLamp. To knock in *vp39* promoter, a fragment containing the *vp39* promoter and *ie0* flanking sequences was amplified via PCR from a synthetic *vp39* promoter mutant template using the primers VP39-F and VP39-R. The amplified fragment was transformed into HS996 competent cells containing the Red®/ET® plasmid pSC101-BAD-gbaA and bacmid BacI-rpsLamp, so that the rpsL-amp cassette could then be removed and replaced by *vp39* promoter, and this generated bacmid was named as BacI-vp39. All primer sequences are listed in Table 1.

To obtain the complete recombinant baculoviruses, plasmid pTriEx-GFP was co-transfected with BacI or BacI-vp39 into *Sf9* cells using FuGENE HD Transfection Reagent (Promega) respectively. Supernatants containing the recombinant baculoviruses were harvested at 96 hours post-transfection (hpt) by centrifugation at 1800 rpm for 5 minutes to remove cell debris, the newly recombined baculoviruses were named vAc-WT or vAc-oeIE01

**Table 1. Primers used in PCR analysis.**

| Name | Primer sequence (5′-3′) |
| --- | --- |
| **AMP-F** | ACGCTCGCTTGCGCGCCGGATAGTATAAGTAATTGATAACGGG CAACGCAACAGGATGGCCTGGTGATGG |
| **AMP-R** | ACGTCATTATATTTTCCTGGACGTTCAGCACGTGACTGCTGGTT CTTATCATTTACCAATGCTTAATCAG |
| **VP39-F** | ACGCTCGCTTGCGCGCCGGATAGTATAAGTAATTGATAACGGG CAACGCAACTCTTGGCTAAATTTATTGAATAAGAG |
| **VP39-R** | ACGTCATTATATTTTCCTGGACGTTCAGCACGTGACTGCTGGTT CTTATCATATTGTTGCCGTCATAAATATG |
| **rpsl-AMP** (identify)-F | AGTTATCTACACGACGGG |
| rpsl-AMP (identify)-R | ATGTCGGCGTTGTACATG |
| vp39 identify-F | TCAGAAAATTGCCGTGGTCC |
| IE0-primerF | ACGTCAAACTGTGCGTCATC |
| IE0-primerR | GCTGGTGTACGACGCGTTAA |
| IE1-primerF | AGCTGTGCAACCCTTGAACA |
| IE1-primerR | GGTCGGAGAACCTGTTGGAA |
| Ecd-primerF | GTCCGGATTCGTTATCATGGGA |
| Ecd-primerR | GCAAAGATGATGAGGCAAATCTGA |

respectively. Viral titers were determined by 50% tissue culture infective doses (TCID50) assay. The approaches for the generation of recombinant baculovirus are depicted in Fig 1. Baculovirus amplification using a multiplicity of infection (MOI) of 0.1 and protein expression occurs at an MOI of 5.

## RT-qPCR quantification of IE0/IE1 expression

To determine the levels of *ie0* and *ie1* mRNA in *Sf*9 cells after inserting the *vp39* promoter, total RNAs were harvested from *Sf*9 cells at 12, 24, 48, 72 hpi respectively and reverse transcribed using the HiScript III cDNA Synthesis Kit (Vazyme Biotech). RT-qPCR detection of *ie0* and *ie1* mRNA was performed using the ChamQ Universal SYBR qPCR Master Mix (Vazyme) according to the manufacturer's instructions. The primers used for detecting *ie0* mRNA are IE0-primerF and IE0-primerR. Similarly, for detecting *ie1* mRNA, the primers are IE1-primerF and IE1-primerR. We selected the *ecd* gene of *Sf*9 cells as an internal control and detected it using the primers Ecd-primerF and Ecd-primerR. To detect the DNA levels of the recombinant virus, we extracted total DNAs from *Sf*9 cells infected with vAc-WT and vAc-oeIE01 using the DNA extraction solution (Rhawn). qPCR analysis was conducted using detection primers for *ie0* and *ie1* mRNA. The reaction was carried out on the BIO-RAD CFX96 Touch, and data processing was done with the software Bio-Rad CFX Maestro. Relative expression of mRNA was normalized to *ecd* RNA using the $2^{-\Delta\Delta Ct}$ method, and data processing was performed by GraphPad.

## Analysis of virus growth curve

To compare the growth curve of vAc-WT and vAc-oeIE01, we infected *Sf*9 cells with the recombinant virus at a MOI of 0.5. The supernatants of infected cell cultures were collected at 24, 48, 72 and 96hpi respectively, and the virus titers were determined using the TCID50 endpoint dilution method. The growth curves were generated using GraphPad software.

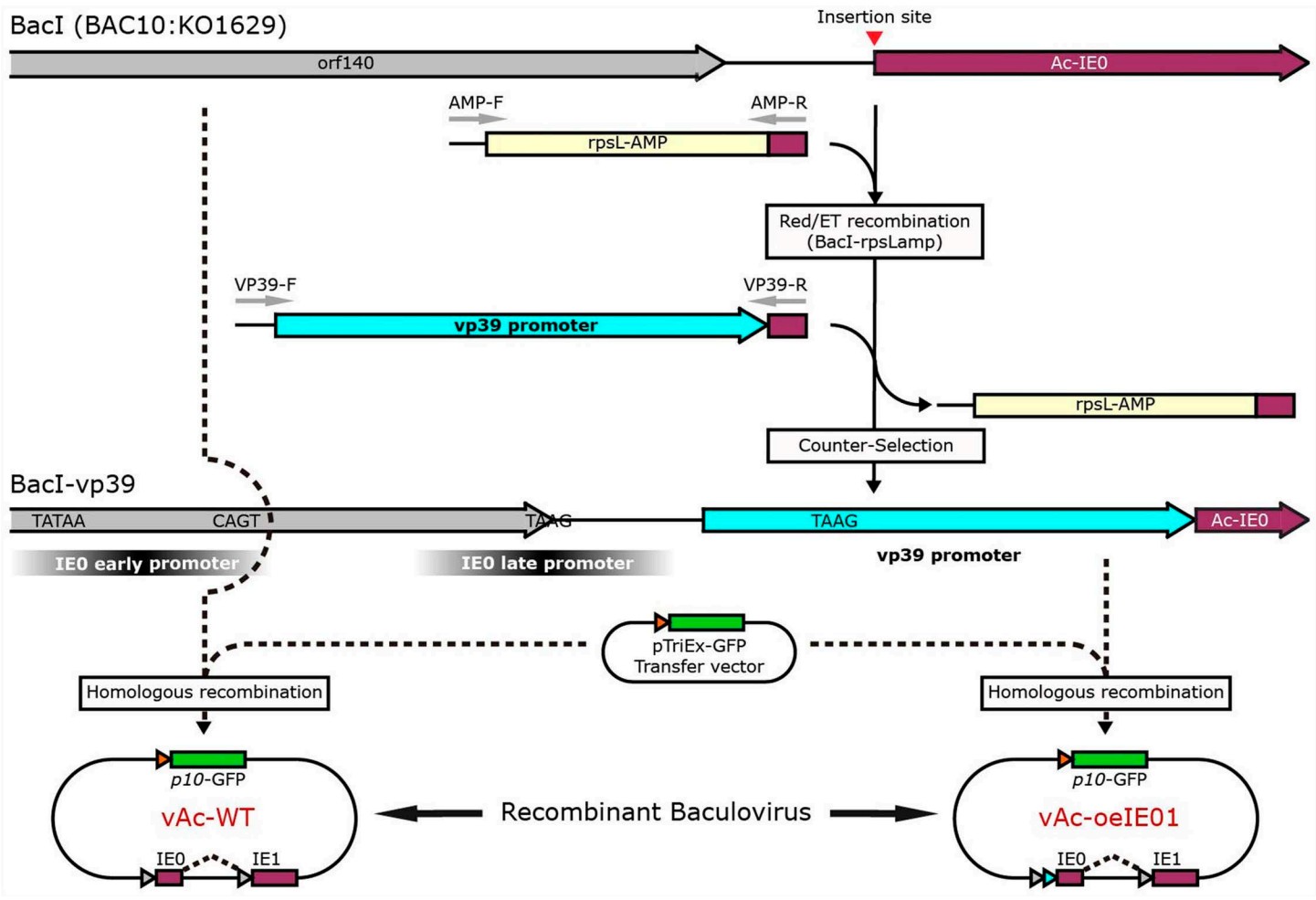

**Fig 1. Model schematic diagram of recombinant baculovirus.**

## Cells apoptosis analysis

*Sf*9 cells were infected with vAc-WT and vAc-oeIE01 at a MOI of 0.5 at 72 hpi, baculovirus-infected cells were harvested by centrifugation at 100 rpm for 5 minutes and the pellets were resuspended in phosphate-buffered saline (PBS). The determination of apoptosis was performed using Annexin V-PE/7-AAD Apoptosis Detection Kit (Vazyme Biotech). Data were collected using NovoCyte Flow Cytometry (Agilent ACEA Biosciences) from at least 10,000 cells. Meanwhile, inverted optical microscope (Leica) is used to observe the baculovirus-infected cells morphological change on the 4 dpi and 5 dpi.

## Determination of protein expression

To quickly detect the expression of GFP (Green fluorescent protein) in cells at 5 dpi, flow cytometry (Agilent ACEA Biosciences) was employed to assess the GFP fluorescence intensity. Subsequently, cells were collected at 72, 96, and 120 hpi to investigate the effect of time on protein expression. To more precisely determine the increase in the proportion of exogenous protein, we optimized the expression conditions and adjusted the loading volume for SDS-PAGE to prevent protein band overloading. Specifically, we opted for suspension culture

in shake flasks. This method was selected because, compared to adherent culture, suspension culture reduces mechanical damage to cells, enhances oxygen supply, minimizes cell aggregation, and provides a more favorable environment for cell growth and protein expression. The *Sf*9 cells in the shake flasks were maintained at a density of $1 \times 10^6$ cells/mL and infected at a MOI of 5. The cells were cultured at 27°C with shaking at 100 rpm for 5 dpi.

Following cell harvest, protein samples were prepared by adding $1 \times$ SDS sample loading buffer and boiling at 100°C for 10 minutes, after which 25 μL per well was loaded. Protein samples were separated by 10% SDS-PAGE and stained with Coomassie Brilliant Blue R-250 for 12 hours. Additionally, the target protein bands were quantified using densitometric scanning with ImageJ software, and the data were analyzed using GraphPad.

### Statistical analysis

The experimental data were presented as the mean ± standard deviation (S.D.) of at least three independent experiments. The level of statistical significance was determined by T-test analysis. P values less than 0.05 were considered significant differences. All data were analyzed using GraphPad Prism 8.0.

## Results

### Insertion of the *vp39* promoter has enhanced the expression of IE0/IE1

To verify the effectiveness of inserting the *vp39* promoter, we conducted RT-qPCR analysis on *Sf*9 cells infected with vAc-WT and vAc-oeIE01. Since *ie0* and *ie1* are early genes of baculovirus involved in regulating the expression of various viral genes, we chose the *ecd* gene from *Sf*9 cells, which exhibits stable expression throughout the entire viral infection cycle, as the internal reference gene [33]. The RT-qPCR results showed that after inserting the *vp39* promoter, the mRNA levels of *ie0* and *ie1* continued to increase from 24 hpi onwards. At 72 hpi, *ie0* and *ie1* mRNA increased by 72.6% and 20%, respectively (Fig 2A, B). This is because the quantification of the *ie1* mRNA included transcription products from the *ie1* promoter, which may reduce the observed differences in quantitative results. Meanwhile, since transcription levels are also correlated with the content of virus template DNA, and overexpression of IE0/IE1 can also enhance the replication level of the viral genome, we further examined the viral genomic DNA in the samples and found that the DNA content increased by 42% (Fig 2C). The increase in IE0 and IE1 transcription levels can be attributed to both the contribution of the *vp39* promoter and the increase in viral DNA template, but these two effects are difficult to separate completely. In any case, our results indicate that the overexpression of IE0/IE1 was achieved using the tandem promoter.

### Overexpressing IE0/IE1 accelerates viral replication

To compare the replication level of vAc-WT and vAc-oeIE01, recombinant baculoviruses were used to infect *Sf*9 cells at a MOI of 0.5 in 12-well plates. Supernatants were collected every 24 hours and their titers were measured until 96 hpi to obtain the growth curve of the baculoviruses (Fig 3). The results showed that compared with vAc-WT, vAc-oeIE01 produced significantly elevated levels of BV from 72 hpi. So, the growth curve confirms that overexpression of IE0/IE1 accelerates the viral infection process and increases the final viral titer.

### Overexpression of IE0/IE1 accelerates cells apoptosis

It is well-known that infection with baculoviruses can induce apoptosis in insect cells. In this study, we collected *Sf*9 cells infected with both vAc-WT and vAc-oeIE01 at 72 hpi. The

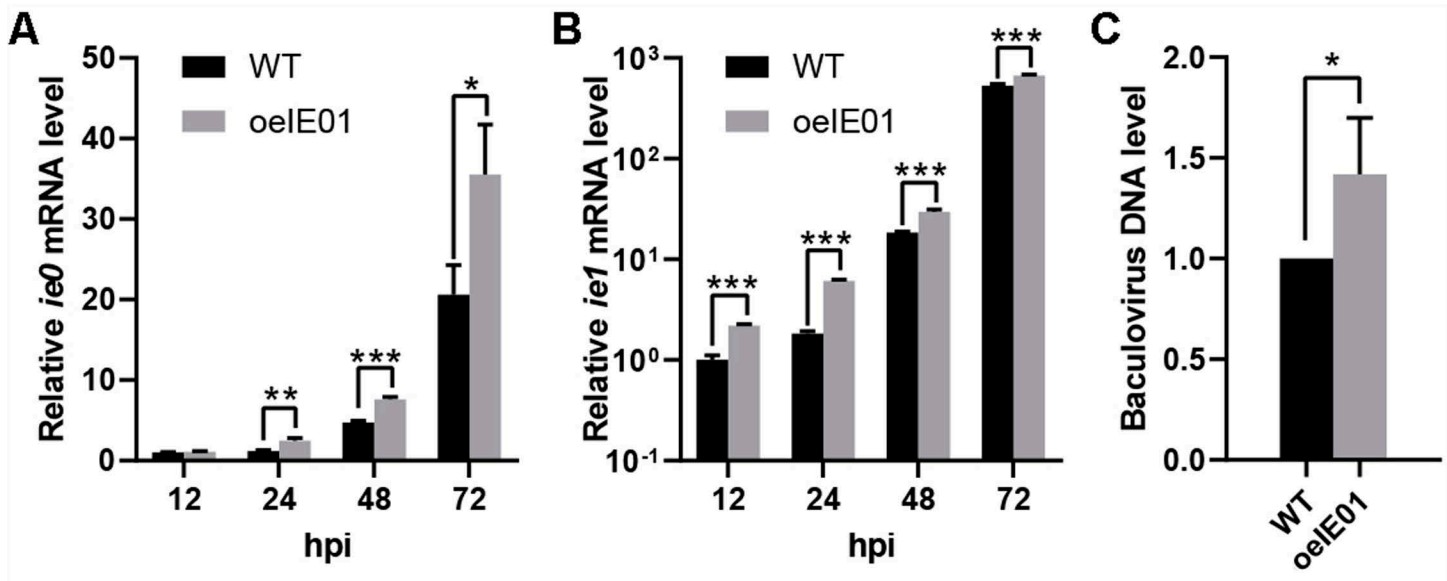

**Fig 2. Insertion of the *vp39* promoter has enhanced the expression of IE0/IE1.** (A) The mRNA level of *ie0* in virus-infected cells were measured at 12-72 hpi. (B) The mRNA level of *ie1* in virus-infected cells were measured at 12-72 hpi. Total RNA samples were extracted from virus-infected cells at 12-72 hpi. The levels of *ie0/ie1* mRNA were determined with RT-qPCR using *ecd* mRNA as the internal control. (C) DNA level of baculovirus. Total DNA samples were prepared from virus infected cells at 72 hpi. The levels of baculovirus DNA were determined with qPCR. Calibrate vAc-oeIE01 using the Ct values of the vAc-WT and calculate the average. (*$p \leq 0.05$; **$p \leq 0.01$; ***$p \leq 0.001$).

apoptotic status of the infected cells was then examined by Annexin V-PE/7-AAD double staining and flow cytometry. The results showed that the percentage of late apoptosis (Q2) in cells infected with vAc-oeIE01 was 6.7% higher than that in the vAc-WT group (Fig 4A). And, in comparison to the control group, the experimental group exhibited a more conspicuous cells apoptosis under optical microscopy, particularly on the 5 dpi (Fig 4B). Taken together, overexpression of IE0/IE1 modestly increased cells apoptosis.

## Overexpression of IE0/IE1 effectively enhances exogenous protein expression

To assess the exogenous gene expression capacity of the new BEVS, we employed the plasmid pTriEx-GFP integrated with BacI-vp39 to construct a recombinant baculovirus. This plasmid contains a *gfp* gene controlled by the *p10* promotor, which serves as a reporter gene for quantifying the amount of exogenous protein expressed. Differences in GFP expression were detected by SDS-PAGE and flow cytometry techniques. We measured protein expression levels for 3-5 dpi and found that in *Sf9* cells infected with vAc-oeIE01 compared to vAc-WT, the production of GFP was higher (Fig 5A). Flow cytometry analysis revealed that the cells infected with vAc-oeIE01 showed a 72% increase in the relative fluorescence intensity of GFP at 5 dpi (Fig 5B). To precisely assess the rise in exogenous protein levels, we re-infected the cells and carefully optimized the conditions for expression, ensuring that the entire process was at its optimal state. Meanwhile, we fine-tuned the protein loading volume to ensure that the GFP band did not over-loading during electrophoresis. Following the completion of protein bands density scanning, separate samples were collected from two independent experimental groups for quantitative analysis. The results demonstrated a significant upregulation of protein expression in cells infected with vAc-oeIE01 compared to the control group, with the increase that exceeded twofold (Fig 5C, D). The above results indicate that infection

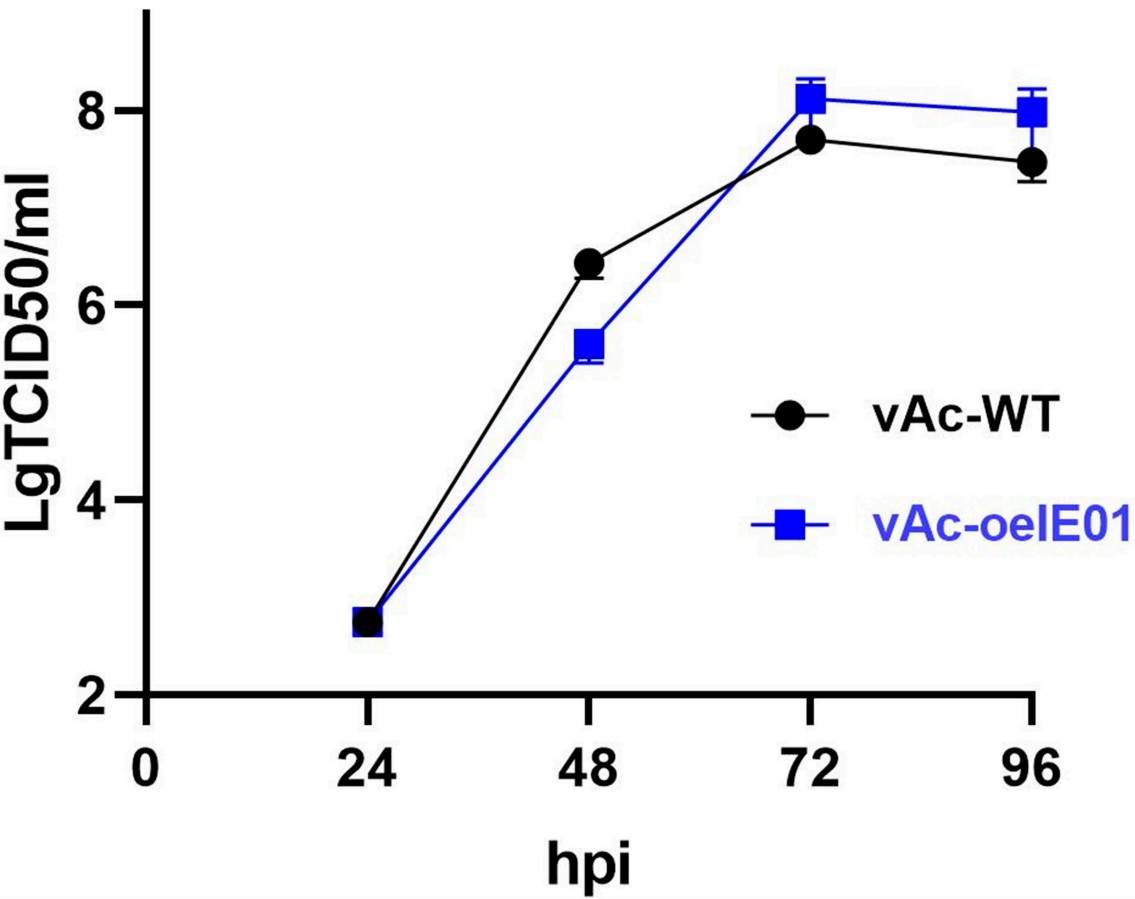

**Fig 3. Virus growth curve.** *Sf*9 cells were infected by vAc-WT and vAc-oeIE01 at a MOI of 0.5. The cell culture supernatants from the infected cells were collected at intervals of 24 hpi, and virus titers were determined by the endpoint dilution assay in triplicate.

with vAc-oeIE01 can significantly enhance the production of exogenous proteins, and these proteins accumulate over time.

## Discussion

As a highly effective insect cell expression system, the BEVS is extensively employed in both laboratory research endeavors and industrial manufacturing processes. Despite its utility, BEVS has been criticized for its relatively low expression capacity compared to other systems, particularly prokaryotic ones. Scientists have employed numerous experimental strategies to optimize exogenous gene expression in the BEVS, but efforts directed towards boosting foreign gene expression through virus regulatory factor modification remain rather limited. Given that viral regulatory factors can control virus growth through either *cis-* or *trans-*acting mechanisms, therefore modulating their expression within the viral genome offers a novel strategy for optimizing the BEVS.

As a multifunctional protein, IE1 plays a pivotal role in early gene activation, DNA replication, and late gene expression. Recent studies have demonstrated that IE1 can stimulate the transcription of viral late promoters independently of viral RNA polymerase [34]. These findings underscore IE1's significance as an expression factor, modulating its expression maybe enhance the expression of late viral genes. In 2014, a study demonstrated that overexpression

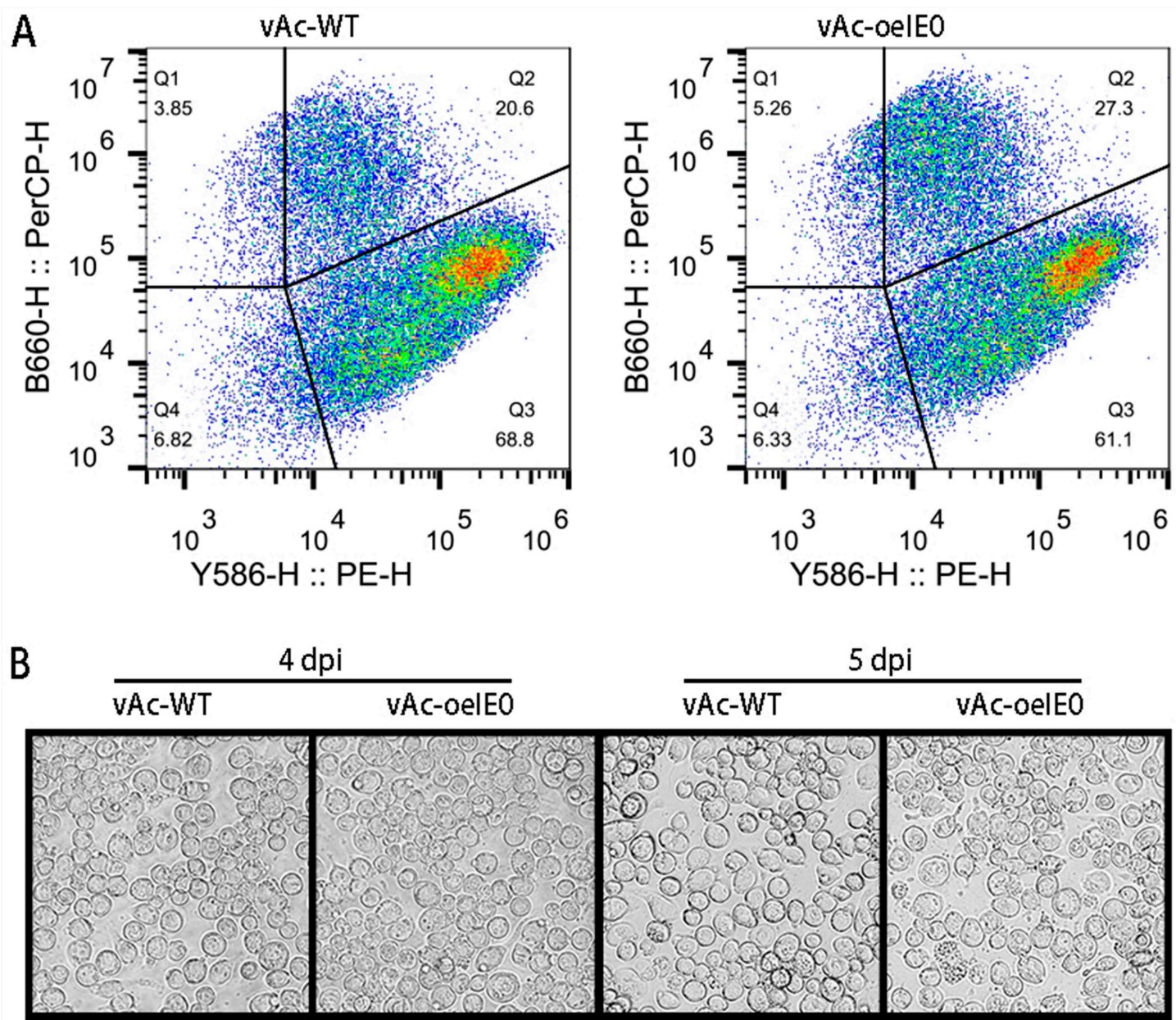

**Fig 4. The overexpression of IE0/IE1 can augment cellular apoptosis.** (A) Apoptosis analysis by flow cytometry. Infected cells were double-stained by Annexin V-PE/7-AAD at 3 dpi, and then proceed with flow cytometry analysis. (B) Cell morphological changes in infected experimental and control groups following viral infection.

of the *ie01* cDNA under the control of the late promoter *polyhedrin* significantly enhanced both cell viability and foreign gene production in the experimental group [35]. In the present study, we employed the *vp39* promoter to overexpress IE01 and observed an increase in cellular apoptosis. Subsequent analysis revealed an upregulation in the mRNA levels of inhibitor of apoptosis proteins (IAPs) within the experimental group (S2 Fig). These findings suggest that IE01, functioning as an earliest pivotal regulatory factor in baculovirus, exerts a global influence on the expression of downstream genes when overexpressed. Given that the expression efficiency of the *vp39* promoter is significantly lower than that of *polyhedrin*, we speculate that the

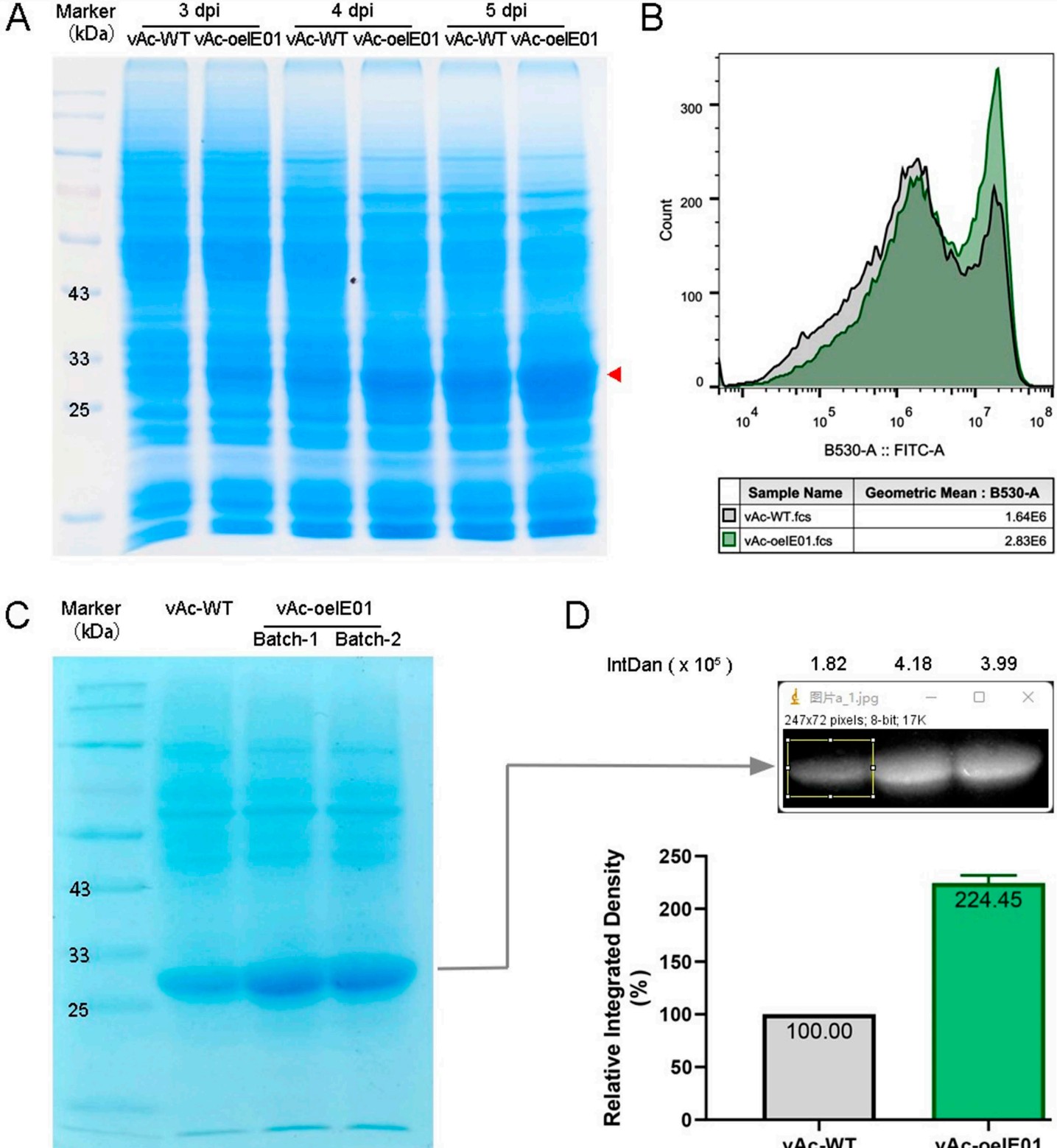

**Fig 5. The production of GFP in *Sf*9 cells.** (A) The relationship between protein production and viral infection time. Total proteins in the cells were separated by SDS-PAGE electrophoresis and visualized by staining with Coomassie Blue R250. The target protein is indicated by a red arrow. (B) Detection of GFP expression level using flow cytometry. *Sf*9 cells infected with virus were collected at 5 dpi, and then subjected to flow cytometry analysis. (C) Detection of the GFP protein levels by SDS-PAGE at 5 dpi. (D) Assessment of protein expression increments. Quantify the GFP expression level of vAc-WT and vAc-oeIE01 by performing density scans, and analyze the data using GraphPad software to present the results.

moderate overexpression of IE01 achieved using the *vp39* promoter may result in insufficient anti-apoptotic activity, thereby failing to fully mitigate the apoptosis induced by overexpression of IE01. Consequently, we observed a slight increase in apoptosis in the experimental group.

Our laboratory previously identified that homologous sequences exceeding 60 bp in the baculovirus genome can mediate significant homologous recombination, leading to defective virus generation and compromising viral genome stability [36]. Given the 1911 nt length of the *ie0/ie1* cDNA, its ectopic expression poses a high risk of homologous recombination. To mitigate this risk, we utilized the native *ie0/ie1* coding sequence within the baculovirus genome for IE0/IE1 overexpression. Specifically, we inserted the *vp39* promoter upstream of the IE0 coding region, ensuring IE0 overexpression during late viral infection without disrupting the native promoter. The *vp39* promoter was chosen for its higher transcriptional activity than the *ie0* promoter but lower than the very late *p10* and *polyhedrin* promoters [37]. This suitable balance enables sufficient IE0/IE1 overexpression without overwhelming the expression of exogenous genes by competing for limited cell resources. Additionally, given comprehensive understanding of the *vp39* promoter, we can introduce point mutations in non-functional regions to avoid homologous recombination with the native *vp39* sequence from the virus [38]. Utilizing tandem promoters, we have attained IE0/IE1 overexpression efficiently, resulting in enhanced viral genome replication and titers. In comparison to Gomez-Sebastián's approach of overexpressing *ie0/ie1* cDNA using the late promoter *polyhedrin* [35], we believe that due to limited late transcription resources, this approach will lead to competitive relationship between *ie01* and exogenous genes, potentially reduces exogenous genes expression unintentionally. Consequently, we employed promoters with lower expression capacity, resulting in improved IE0/IE1 expression levels and noteworthy results. As our ultimate objective, the expression level of GFP controlled by the *p10* promoter was significantly increased, as evidenced by SDS-PAGE and Western blot analyses (S1 Fig). This suggests that overexpressing IE0/IE1 through tandem promoter to improve BEVS expression capacity is a feasible approach.

In our research, without the use of *hr* enhancer, we only overexpressed IE0/IE1 to explore its impact on exogenous genes governed by late promoters. Unexpectedly, our findings showed that IE0/IE1 could directly regulate late promoter activity without the involvement of *hr* enhancer. In fact, our previous research has found that *hr*, at least hr4a, can only enhance the activity of the early promoter of the baculovirus, rather than the late promoter [36].

Our previous research successfully knocked down the *caspase-1* in insect host cells using RNA interference technology and constructed a baculovirus expression vector with anti-apoptotic effects [13]. In the future, combining these strategies with IE0/IE1 overexpression maybe mitigate the negative impact on cell apoptosis.

## Conclusion

We successfully constructed a novel baculovirus expression vector by overexpressing IE01 while avoiding the risk of homologous recombination. This vector can increase the expression level of exogenous genes by approximately two-fold. Furthermore, this study provides new insights for future optimization of the BEVS through the regulation of viral regulatory factors.

## Supporting information

**S1 Fig. Western blot analysis of GFP expression in *sf*9 cells.** Infect s*f*9 cells with vAc-WT and vAc-oeIE01 at an MOI of 5 for 5 dpi. GFP was detected through Western blot using a GFP-specific antibody and an HRP-conjugated Goat Anti-Rabbit antibody were used as secondary antibody. Quantification of GFP signals of vAc-WT and vAc-oeIE01 by performing density scans, and ImageJ software was used to analyze and present the results. (JPG)

**S2 Fig. Insertion of the *vp39* promoter has enhanced the expression of IAPs.** (A) The mRNA level of *IAP1* in virus-infected cells were measured at 12-48 hpi. (B) The mRNA level of *IAP2* in virus-infected cells were measured at 12-48 hpi. Total RNA samples were extracted from virus-infected cells at 12-48 hpi. The levels of *IAP1/IAP2* mRNA were determined with RT-qPCR using *ecd* mRNA as the internal control. The following primers were used for RT-qPCR: IAP1: IAP1-U (GCAAAGTCTGTCTCGAACGC) and IAP1-D (ACGACACGTC-GGACACTTTT); IAP2: IAP2-U (GCCGGCACAAACAAAATTGC) and IAP2-D (AGGAAT-CAAATCGGCAGCCA). (**$p \leq 0.01$; ***$p \leq 0.001$).
(JPG)

**S1 Raw image. Raw image.**
(PDF)

## Acknowledgements

We are grateful for the assistance provided by Dr. Ningjuan Fan from the Teaching and Research Core Facility at College of Life Sciences, Northwest A&F University in the area of experimental instruments. We also appreciate the guidance from Ms. Lan Lan and Ms Yang Zhao in experimental techniques. Additionally, we thank the anonymous reviewers for their valuable feedback on the early versions of this manuscript.

## Author contributions

**Data curation:** Xiaodong Xu, Sijun He, Weining Li, Ruirui Zhang.

**Formal analysis:** Wangcheng Song.

**Project administration:** Xiaodong Xu.

**Software:** Sijun He.

**Supervision:** Hao Nan.

**Writing – original draft:** Sijun He.

**Writing – review & editing:** Xiaodong Xu.

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
