## [Decision Letter · Decision Letter 0]

13 Dec 2024

PONE-D-24-48349Improving the production of baculovirus expression vector by overexpression of IE0/IE1 through tandem promoterPLOS ONE

Dear Dr. Xu,

Thank you for submitting your manuscript to PLOS ONE. After careful consideration, we feel that it has merit but does not fully meet PLOS ONE’s publication criteria as it currently stands. Therefore, we invite you to submit a revised version of the manuscript that addresses the points raised during the review process. Please note the reviewers’ comments indicating both major and minor concerns regarding your manuscript. Specifically, the focus is on the expression of IE1/IE0 under the promoters used in this research and the consequences after the adjusted expressions. In addition, I strongly recommend that the authors confirm protein expression of the proteins (IE1/IE0, IAPs, etc) of interest using Western blotting or alternative techniques. The authors must avoid overstating and exaggerating their conclusions to match the study’s limitations. Careful consideration must be given to the style and language used in this round revision.

We look forward to receiving your revised manuscript.

Kind regards,

Jian Xu, Ph.D.

Academic Editor

PLOS ONE

“This work was supported by Key Research and Development Program of Shaanxi (Program No. 2022QCY-LL-52) and a collaborative project with Shaanxi Sky Pet Biotechnology Co., Ltd (Program No. K4050722064).”

Reviewers' comments:

Reviewer's Responses to Questions

**Comments to the Author**

1. Is the manuscript technically sound, and do the data support the conclusions?

Reviewer #1: Yes

2. Has the statistical analysis been performed appropriately and rigorously? 

Reviewer #1: No

3. Have the authors made all data underlying the findings in their manuscript fully available?

Reviewer #1: Yes

4. Is the manuscript presented in an intelligible fashion and written in standard English?

Reviewer #1: Yes

5. Review Comments to the Author

Reviewer #1: This study adopted a strategy to enhance the expression of the IE01 gene to improve the baculovirus expression vector system. Instead of introducing an additional ORF sequence for IE01, the authors implemented an approach that involved adding a late promoter to the virus genome's native IE01 gene, complementing its original promoter. This method successfully reduced the risk of homologous recombination and increased the yield of target protein expression. The paper provides interesting data but it still needs a revision to be acceptable for the journal.

Major comments

The authors quantified the mRNA levels of the ie0 and ie1 genes in vAc-oeIE01 at 72 hpi, but how do their expression levels change at other time points? Including time-course data could provide valuable insights into when IE01 expression is most effective and its potential functional significance.

In this study, which of the two, IE0 or IE1, do the authors consider to be the primary contributor to the observed improvements in viral replication and target protein expression levels?

As noted in the Introduction, strategies that prolong cell survival post-infection are generally effective for enhancing target protein production. However, the current results indicate an increase in apoptosis, which contrasts with this principle. Could the authors provide their perspective on why protein expression levels improved despite this increase in apoptosis? Additionally, is 5 dpi the optimal time point for protein expression? The results suggest that the peak in expression may occur earlier due to accelerated viral replication. Would it be possible that at 6 dpi or 7 dpi, the expression levels of the WT catch up to those observed in the current experiment?

As the authors mentioned, previous studies have reported that overexpression of IE01 under the polyhedrin promoter enhances cell survival post-infection. In contrast, the VP39 promoter used in this study appears to promote apoptosis. Could this be due to insufficient levels of IE01 to support cell survival, with suboptimal expression levels triggering cell death? This observation raises the question of whether achieving an optimal level of IE01 expression is critical for balancing cell survival and apoptosis in this context.

Minor comments

The reference in the first sentence of the Introduction is numbered as 28. Please ensure that the references are renumbered in the correct sequential order throughout the manuscript.

Methods and Materials

→Materials and Methods

There is no description of statistical analysis in the text. Please provide details in Materials and Methods.

Please mention error bars in Legends for some Figures, e.g., standard deviation or standard error.

The authors stated in the results section that ‘To detect the increased proportion of exogenous proteins more accurately, we optimised the expression conditions and adjusted the loading volume of SDS-PAGE to avoid overloading of protein bands.’ Please describe under what conditions the optimisation of the expression conditions was carried out.

In the Figure 5, the authors used the p10 promoter for GFP expression instead of the polyhedrin promoter. Could the authors explain the rationale behind this choice?

In this study, GFP was selected as the target to investigate the enhancement of protein expression. Do the authors have any data showing whether the observed increase in expression also applies to other proteins, particularly secreted proteins?

The Financial Disclosure section states that no funding was provided for this study. However, the Acknowledgments mention a program that appears to have supported the research. Could the authors clarify whether funding was received from this program and, if so, ensure consistency between the Financial Disclosure and Acknowledgments sections?

6. PLOS authors have the option to publish the peer review history of their article (what does this mean? ). If published, this will include your full peer review and any attached files.

**Do you want your identity to be public for this peer review?** For information about this choice, including consent withdrawal, please see our Privacy Policy .

Reviewer #1: No

---

## [Author Response · Author response to Decision Letter 1]

5 Jan 2025

Dear Editors and Reviewers:

Thank you for your letter and for the Referees’ comments concerning our manuscript entitled “Improving the production of baculovirus expression vector by overexpression of IE0/IE1 through tandem promoter” (PONE-D-24-48349). Those comments are all valuable and very helpful for revising and improving our paper, as well as the important guiding significance to our further researches. We have carefully studyed these comments and made corrections to the manuscript, hoping for your approval. Meanwhile, we have thoroughly considered each comment and made every effort to address them.

Thank you for suggesting that we provide Western-blot results to confirm the increase in target protein expression. As shown in Figure S1, sf9 cells were infected with vAc-WT and vAc-oeIE01 at an MOI of 5, and cells were collected at 5 dpi for electrophoresis detection. Following Western-blot analysis, we confirmed that the expression level of the exogenous protein in the experimental group increased approximately two-fold.

S1 Fig. Western Blot Analysis of GFP Expression in sf9 cells. Infect sf9 cells with vAc-WT and vAc-oeIE01 at an MOI of 5 for 5 dpi. GFP was detected through Western blot using a GFP-specific antibody and an HRP-conjugated Goat Anti-Rabbit antibody were used as secondary antibody. Quantification of GFP signals of vAc-WT and vAc-oeIE01 by performing density scans, and ImageJ software was used to analyze and present the results.

Reviewer #1:

Major comments

1.Response to comment: “The authors quantified the mRNA levels of the ie0 and ie1 genes in vAc-oeIE01 at 72 hpi, but how do their expression levels change at other time points? Including time-course data could provide valuable insights into when IE01 expression is most effective and its potential functional significance”.

Response: Thank you for your question. According to the updated experimental results（Fig 2A, B）, the transcription levels of both ie0 and ie1 in the experimental group showed a continuous increase starting from 24 hpi compared to the control group. This upward trend can be attributed to the insertion of the vp39 promoter. Therefore, our findings demonstrated that the insertion of the vp39 promoter effectively enhances the expression of IE01, and this enhancement can persist into the late stages of viral infection.

2. Response to comment: “In this study, which of the two, IE0 or IE1, do the authors consider to be the primary contributor to the observed improvements in viral replication and target protein expression levels?”

Response: Thank you for your question. Currently, we cannot determine which one, IE0 or IE1, is the primary contributor to the increased expression of exogenous proteins. A 2004 study, showed that IE0 is expressed at a higher level than IE1 during the early stages of viral infection, but its expression level decreased while IE1’s gradually increases in the later stages [1]. This study also found that high-level co-expression of IE0 and IE1 antagonizes the activation of late genes, and their co-expression in a specific ratio is required for maximal activation of very late genes[1] A 2014 study indicated that while the individual expression of IE0 or IE1 can promote viral DNA replication, their co-expression is far more effective [2]. Therefore, we speculate that both IE0 and IE1 play a trans-activation role, which is most effective when they are co-expressed in a certain ratio. Using the vp39 promoter to overexpress both IE0 and IE1, we anticipated that their naturally ratio of IE0/IE1 would enhance the activation of very late promoters, which was indeed confirmed.

3. Response to comment: “As noted in the Introduction, strategies that prolong cell survival post-infection are generally effective for enhancing target protein production. However, the current results indicate an increase in apoptosis, which contrasts with this principle. Could the authors provide their perspective on why protein expression levels improved despite this increase in apoptosis? Additionally, is 5 dpi the optimal time point for protein expression? The results suggest that the peak in expression may occur earlier due to accelerated viral replication. Would it be possible that at 6 dpi or 7 dpi, the expression levels of the WT catch up to those observed in the current experiment?”

Response: Thank you for your question. This is a very good question, and I will answer it one by one for you. 1. Apoptosis has a negative impact on the expression of exogenous proteins, while overexpression of IE01 promotes viral DNA replication, resulting in a positive effect on exogenous protein expression. According to our experimental results, compared to the WT, apoptosis increased by only about 7% after IE01 overexpression, which is not significant. Therefore, we believe that the positive effect of overexpressing IE01 can counteract the negative impact of apoptosis, leading to an overall increase in exogenous proteins expression. 2. Normally, the expression timeframe of exogenous proteins in the baculovirus insect cell expression system is not exceed 4 days [3]. However, we found that although apoptosis was obvious at 5 dpi, total expression of exogenous protein still increased compared to 4 dpi due to protein accumulated in the cells. Therefore, we presented the data for both 4 dpi and 5 dpi, but these results do not indicate that 5 dpi is the optimal time point for exogenous protein expression. Instead, these results suggest that 5 dpi may yield better expression levels in specific cases. 3. Protein expression in insect cells does not extend beyond 6 dpi, as prolonged infection causes host cell lysis, releasing proteases that degrade the target protein. Our laboratory has constructed an anti-apoptotic baculovirus expression vector, but even with this, exogenous protein expression only lasts until 5 dpi [4].

4. Response to comment: “As the authors mentioned, previous studies have reported that overexpression of IE01 under the polyhedrin promoter enhances cell survival post-infection. In contrast, the VP39 promoter used in this study appears to promote apoptosis. Could this be due to insufficient levels of IE01 to support cell survival, with suboptimal expression levels triggering cell death? This observation raises the question of whether achieving an optimal level of IE01 expression is critical for balancing cell survival and apoptosis in this context.”

Response: Thank you for your question. As you mentioned, the inspiration for our article came from a paper published in 2014 [5]. The role of IE0 is still unclear, and it is currently speculated to stimulate the transcription of IE1 early in infection, accelerating the synthesis of IE1 and promoting viral DNA replication. While IE1 is known to induce apoptosis, it can counteract this by upregulating anti-apoptotic factors [6-7]. Currently, we are not sure which of IE0/IE1 play dominated role. Based on our current experimental results, we tend to agree with your viewpoint. The expression capacity of the vp39 promoter we used is much lower than that of the very late promoter [8]. The expression level of IE01 likely isn’t sufficient for cell survival, triggering apoptosis. Therefore, optimizing the expression level of IE01 is crucial for balancing cell survival and apoptosis. Compared to cells infected with the WT virus, the experimental group shows only slight increase in apoptosis. Our ultimate goal is to construct a baculovirus expression vector that enhances exogenous proteins expression, so most of the late transcriptional resources need to be devoted to the expression of exogenous proteins. Gómez-Sebastián's use of the polyhedrin promoter to overexpress IE01 may waste valuable late-phase transcription resources. We have found that the overexpression of IE01 using the vp39 promoter obviously increases the level of exogenous protein expression, and mild apoptosis does not seem to impact our goal.

Minor comments

1.Response to comment: “The reference in the first sentence of the Introduction is numbered as 28. Please ensure that the references are renumbered in the correct sequential order throughout the manuscript.”

Response: Thank you very much for pointing out the formatting issue. We have revised and organized the references according to the requirements of the journal, and you can see them in the revised manuscript we submitted.

2. Response to comment: “There is no description of statistical analysis in the text. Please provide details in Materials and Methods. Please mention error bars in Legends for some Figures, e.g., standard deviation or standard error.”

Response: Thank you very much for pointing out the formatting issue. The experimental data were presented as the mean ± standard deviation (S.D.) of at least three independent experiments. The level of statistical significance was determined by T-test analysis. P values less than 0.05 were considered significant differences. All data were analyzed using GraphPad Prism 8.0. We have added the description of these statistical analyses to the Methods and Materials section of the revised manuscript.

3. Response to comment: “The authors stated in the results section that ‘To detect the increased proportion of exogenous proteins more accurately, we optimised the expression conditions and adjusted the loading volume of SDS-PAGE to avoid overloading of protein bands.’ Please describe under what conditions the optimisation of the expression conditions was carried out.”

Response: Thank you for your question. In previous experiments, we used a six-well plate for adherent cell culture. In this experiment, we opted for suspension culture in shake flasks. This is because compared to adherent culture, suspension culture can reduce cell mechanical damage, increase oxygen supply to cells, reduce cell aggregation, and is more suitable for cell growth and protein expression. The density of sf9 cells in shake flasks is 1×10^6 cells/mL, with virus infecting cells at an MOI of 5, cultured in an oscillating incubator at 27°C and 100 rpm until the 5dpi. After cells collection, we added 1×SDS sample loading buffer, boiled it at 100°C for 10 minutes to prepare protein samples, and then performed SDS-PAGE.

4. Response to comment:“In the Figure 5, the authors used the p10 promoter for GFP expression instead of the polyhedrin promoter. Could the authors explain the rationale behind this choice?”

Response: Thank you for your question. The p10 promoter and the polyhedrin promoter are both strong late-stage promoters widely used for the expression of exogenous proteins. Through our experiments conducted over the past year, we observed difference in the expression levels of exogenous proteins when employing these two promoters. The expression level of exogenous proteins using the polyhedrin promoter is lower than that using the p10 promoter. Currently, the reasons for these expression differences are unclear, and we have designed related experiments to investigate this issue. Therefore, for maximal protein production in the experiments, we chose the p10 promoter for protein expression.

5.Response to comment: “In this study, GFP was selected as the target to investigate the enhancement of protein expression. Do the authors have any data showing whether the observed increase in expression also applies to other proteins, particularly secreted proteins?”

Response: Thank you for your inquiry. In this study, we focused on enhancing the expression of GFP and did not conduct experiments on other proteins, especially secretory proteins. After infecting insect cells with baculovirus, the virus hijacks the host's protein translation and transport systems to express its own proteins. We consider that overexpression of IE0/IE1 only affects mRNA levels, without impacting protein translation and translocation. Our previous studies have shown that optimizing the baculovirus expression vector can simultaneously increase the expression levels of intracellular and secreted proteins [4].

6.Response to comment: “The Financial Disclosure section states that no funding was provided for this study. However, the Acknowledgments mention a program that appears to have supported the research. Could the authors clarify whether funding was received from this program and, if so, ensure consistency between the Financial Disclosure and Acknowledgments sections?”

Response: Thank you for pointing out the issues. Yes, this work was supported by the Shaanxi Province Key Research and Development Program (Project No. 2022QCY-LL-52) and a collaborative project with Shaanxi Sky Pet Biotechnology Co., Ltd. (Project No. K4050722064). We have removed this content from the acknowledgments. Please help us publish this funding information in the funding statement section of the online submission form.

References

1. Huijskens, I., Li, L., Willis, L. G., & Theilmann, D. A. (2004). Role of AcMNPV IE0 in baculovirus very late gene activation. Virology, 323(1), 120-130.

2. Sokal, N., Nie, Y., Willis, L. G., Yamagishi, J., Blissard, G. W., Rheault, M. R., & Theilmann, D. A. (2014). Defining the roles of the baculovirus regulatory proteins IE0 and IE1 in genome replication and early gene transactivation. Virology, 468-470, 160-171.

3. Thermo Fisher Scientific. (2019). Bac-to-Bac™ baculovirus expression system user guide: An efficient site-specific transposition system to generate baculovirus for high-level expression of recombinant proteins. https://assets.thermofisher.com/TFS-Assets/LSG/manuals/MAN0000414_BactoBacExpressionSystem_UG.pdf

4. Zhang, X., Zhao, K., Lan, L., Shi, N., Nan, H., Shi, Y., Xu, X., & Chen, H. (2021). Improvement of protein production by engineering a novel antiapoptotic baculovirus vector to suppress the expression of Sf-caspase-1 and Tn-caspase-1. Biotechnology and Bioengineering, 118(8), 2977-2989.

5. Gómez-Sebastián, S., López-Vidal, J., & Escribano, J. M. (2014). Significant productivity improvement of the baculovirus expression vector system by engineering a novel expression cassette. PLoS One, 9(5), e96562.

6. Prikhod'ko, E. A., & Miller, L. K. (1996). Induction of apoptosis by baculovirus transactivator IE1. Journal of Virology, 70(10), 7116-7124.

7. Schultz, K. L., Wetter, J. A., Fiore, D. C., & Friesen, P. D. (2009). Transactivator IE1 is required for baculovirus early replication events that trigger apoptosis in permissive and nonpermissive cells. Journal of Virology, 83(1), 262-272.

8. Chen, Y. R., Zhong, S., Fei, Z., Hashimoto, Y., Xiang, J. Z., Zhang, S., & Blissard, G. W. (2013). The transcriptome of the baculovirus Autographa californica multiple nucleopolyhedrovirus in Trichoplusia ni cells. Journal of Virology, 87(11), 6391-6405.

---

## [Decision Letter · Decision Letter 1]

21 Jan 2025

PONE-D-24-48349R1Improving the production of baculovirus expression vector by overexpression of IE0/IE1 through tandem promoterPLOS ONE

Dear Dr. Xu,

Thank you for submitting your manuscript to PLOS ONE. After careful consideration, we feel that it has merit but does not fully meet PLOS ONE’s publication criteria as it currently stands. Therefore, we invite you to submit a revised version of the manuscript that addresses the points raised during the review process.

We look forward to receiving your revised manuscript.

Kind regards,

Jian Xu, Ph.D.

Academic Editor

PLOS ONE

Journal Requirements:

Reviewers' comments:

Reviewer's Responses to Questions

**Comments to the Author**

1. If the authors have adequately addressed your comments raised in a previous round of review and you feel that this manuscript is now acceptable for publication, you may indicate that here to bypass the “Comments to the Author” section, enter your conflict of interest statement in the “Confidential to Editor” section, and submit your "Accept" recommendation.

Reviewer #1: (No Response)

2. Is the manuscript technically sound, and do the data support the conclusions?

Reviewer #1: Partly

3. Has the statistical analysis been performed appropriately and rigorously? 

Reviewer #1: Yes

4. Have the authors made all data underlying the findings in their manuscript fully available?

Reviewer #1: Yes

5. Is the manuscript presented in an intelligible fashion and written in standard English?

Reviewer #1: Yes

6. Review Comments to the Author

Reviewer #1: While the manuscript has adequately addressed most of the requested points, the following comments must be addressed for acceptance.

Although the Western blot results for GFP (Fig. S1) support the notion of enhanced expression of the target protein, the Editor has specifically requested Western blot analysis of ie0, ie1, and IAP to enable comparison of their protein-level expression. Since increases in mRNA levels do not always correlate with increases in protein abundance, it is important to verify this point.

Additionally, please note that variations in Western blot band signal intensity do not necessarily reflect proportional differences in protein quantity. In other words, a two-fold increase in signal does not necessarily indicate a two-fold increase in the amount of protein. In contrast, the differences shown by the CBB staining in this study are likely more reflective of the actual differences in protein levels.

Minor Comments

Methods and Materials

→Materials and Methods

In the Figure 2, the figure legend does not include an explanation for the three asterisks (***).

Conclussion → Conclusion

Please carefully review the spelling of words throughout the manuscript once again.

Please incorporate the optimized culture conditions described in the Response to Reviewers into the Materials and Methods section.

The Figure S1 is not mentioned in the manuscript.

7. PLOS authors have the option to publish the peer review history of their article (what does this mean? ). If published, this will include your full peer review and any attached files.

**Do you want your identity to be public for this peer review?** For information about this choice, including consent withdrawal, please see our Privacy Policy .

Reviewer #1: No

---

## [Author Response · Author response to Decision Letter 2]

13 Feb 2025

Dear Editors and Reviewers:

We sincerely thank the reviewer for the careful review and valuable comments on our study. The suggestion to perform Western blot analysis of IE0, IE1, and IAPs to compare their protein expression levels is highly important. We fully agree that an increase in mRNA levels does not always correlate with an increase in protein abundance, and thus, it is essential to validate protein expression levels.

However, due to the lack of commercially available specific antibodies against IE0, IE1, and IAPs, we are currently unable to directly perform Western blot analysis for these proteins. Nevertheless, to address the reviewer’s suggestion as much as possible, we detected the transcription levels of IAP1 and IAP2 in both experimental and control groups within 12-48 hpi (Fig S2). The results demonstrated that overexpression of IE01 increased the transcription levels of IAP1 and IAP2. As mentioned in our manuscript, IE01, as an earliest pivotal regulatory factor of the baculovirus, exerts a global influence on the expression levels of downstream genes when overexpressed. Consequently, the observed enhance in the expression levels of IAP1 and IAP2 aligns with our expectations. In line with your previous conjecture, the expression capacity of the vp39 promoter we adopted is significantly lower compared with that of the late promoters p10 and polh. Although the elevated levels of IAPs enhance the anti-apoptotic capacity of cells, this enhancement is insufficient to fully counteract the apoptotic effects triggered by overexpression of IE01.

Meanwhile, we also agree with your opinion that changes in the signal intensity of Western blot bands do not necessarily directly reflect proportional differences in the quantity of proteins. The supplementary Fig.S1 we provided previously was mainly intended to further enrich the data support for the increase in protein expression levels. We will carefully adjust the relevant expressions in the article to ensure the accuracy and rigor of the wording.

We have made appropriate revisions to the other detailed questions you raised about the manuscript. Once again, we sincerely appreciate the insightful comments you provided, which have been of great help during the revision process of our manuscript and significantly improved its quality.

This study was financially supported by two grants:

1. Shaanxi Province Key Research and Development Program (Project No. 2022QCY-LL-52), recipient: Hao Nan

2. A collaborative project with Shaanxi Sky Pet Biotechnology Co., Ltd. (Project No. K4050722064), recipient: Wangcheng Song

We have completed the required revisions in the Cover Letter and re-uploaded the documents accordingly. We kindly request your assistance in updating the funding information in the online submission system.

---

## [Editor Report · Decision Letter 2]

17 Feb 2025

Improving the production of baculovirus expression vector by overexpression of IE0/IE1 through tandem promoter

PONE-D-24-48349R2

Dear Dr. Xu,

We’re pleased to inform you that your manuscript has been judged scientifically suitable for publication and will be formally accepted for publication once it meets all outstanding technical requirements.

Kind regards,

Jian Xu, Ph.D.

Academic Editor

PLOS ONE
---

## [Editor Report · Acceptance letter]

PONE-D-24-48349R2

PLOS ONE

Dear Dr. Xu,

I'm pleased to inform you that your manuscript has been deemed suitable for publication in PLOS ONE. Congratulations! Your manuscript is now being handed over to our production team.

Kind regards,

on behalf of

Dr. Jian Xu

Academic Editor

PLOS ONE